# Preparation of Citral Compound and Its Bamboo Antimildew Properties

**DOI:** 10.3390/polym14214691

**Published:** 2022-11-03

**Authors:** Yingying Shan, Shiqin Chen, Jingjing Zhang, Chungui Du, Chunlin Liu, Fei Yang, Wenxiu Yin, Yuran Shao, Yuting Wang

**Affiliations:** College of Chemistry and Materials Engineering, Zhejiang A&F University, Hangzhou 311300, China

**Keywords:** citral compound, cinnamaldehyde, thymol, bamboo, mold, mildew resistance

## Abstract

To reduce the amount of citral used without reducing the antimildew performance of bamboo, the citral compound preparation process, the distribution of the compound in bamboo, and its antimildew performance were investigated using the Oxford cup method, Fourier-transform infrared spectroscopy, and ultraviolet spectrophotometry. The results revealed that the combination of citral with cinnamaldehyde or thymol may lead to partial chemical reactions, which may change the chemical structure of citral and affect its bacteriostatic properties. The bacteriostatic properties of the citraldehyde thymol compound against common molds of bamboo were considerably superior to those of the citral cinnamaldehyde compound. The limonaldehyde thymol compound showed a low distribution trend outside and vice versa inside in the treated bamboo. The citral thymol compound exhibited good antimildew performance at a concentration of 200 mg/mL. The citral thymol compound could reduce the amount of citral by approximately 67 mg/mL without reducing the antimildew performance of bamboo.

## 1. Introduction

Bamboo, a natural material, has a tensile strength comparable to that of low-carbon steel and good toughness and degradability, and it is highly advantageous in terms of structure and cost [1]. Therefore, bamboo and its products have been widely used for applications such as decoration, architecture, and gardening [2,3,4,5] and exhibit a rapid development trend. However, bamboo and its products are very susceptible to mildew and severe surface pollution; thus, they lose their use value [6,7] and are discarded, which results in considerable wastage of resources and economic losses. Therefore, vigorous bamboo mold prevention research is of great significance.

An antimildew agent for bamboo has recently become a hot research topic, and the research and application of natural antibacterial agents for bamboo mold prevention have been increasing. Citral is one such natural antibacterial agent and is mainly derived from the essential oil of the Chinese medicine Shancangzi. It is a fat-soluble natural mixture with rich fragrance [8] and has received more attention because of its strong broad-spectrum bacteriostatic properties [9,10]. Studies have shown that citral has good antibacterial properties against Staphylococcus aureus, Escherichia coli, Penicillium, and Mucormycosis [11]; citral and its derivatives have a certain inhibitory effect on Camellia oleifera anthracnose, and the inhibitory effect increases with the increase in concentration [12]. Xia [13] studied the inhibitory effect of citral on Aspergillus flavus, the source of Chinese herbal medicines, and the results showed that citral had a good control effect on mildew in Chinese herbal medicines, and the concentrations of Ze diarrhea and citrus aurantium mildew were 4 μL/g and 8 μL/g, respectively. Tao [14,15] studied the antifungal activity and antifungal mechanism of citrus essential oil and citral against Penicillium italicans and Penicillium dactylidae, and found that citral can significantly change the morphology of Penicillium italicans mycelium by causing cytoplasmic loss and deformation of hyphae. However, no one has studied whether citral has an inhibitory effect on bamboo mold, so the authors [16,17] conducted a study on the bacteriostatic performance of citral on common molds in bamboo and its control effect on bamboo treatment. The results showed that, when the citral concentration was 75 mg/mL, the bacteriostatic rate of citral on bamboo molds exceeded 100%. However, when the citral concentration reached 200 mg/mL the control effectiveness of citral against the bamboo mildews reached 100%. The good antimildew performance of citral can be achieved only by increasing the citral concentration. However, when the amount of citral is high, the lemon flavor is strong, which has a negative impact on the sensory characteristics of the product itself and its odor may cause discomfort to consumers. Thus, reducing the amount of citral in bamboo mold prevention is essential. However, this would reduce the antimildew performance of citral, which necessitates studies on the process and method of reducing the amount of citral without reducing its antimildew performance.

We found [16] that the citral treatment of bamboo had a poor control effect on Penicillium citrinum but a better control effect on Aspergillus niger, Trichoderma viride molds, and mixed molds. Studies have shown that the natural antibacterial agent cinnamaldehyde has a cinnamon smell and sweetness, is the main ingredient in the traditional Chinese medicines of cinnamon branch and cinnamon [18], and has broad-spectrum antifungal effects [19,20,21,22,23]. Thymol has the spicy and herbaceous aromas of thyme, is the main component of plant volatile oils (about 50%) [24,25,26,27], and has a broad-spectrum antifungal effect [28,29,30]. Both thymol and cinnamaldehyde are listed as “generally recognized as safe” substances (GRAS) by the US Food and Drug Administration [31,32,33], and can be used as green and safe natural antibacterial agents in food preservation, pharmaceutical preparations, and feed preservatives; in addition, studies have shown that cinnamaldehyde and thymol [34,35,36] have a good antibacterial effect on Penicillium orange. Therefore, in view of the characteristics of cinnamaldehyde and thymol having the characteristics of green, safe, and good antibacterial effect on Penicillium orange, the author selected cinnamaldehyde and thymol as the compound agents of citral for compounding; through the synergistic effect between them, they can reduce the amount of citral use, do not reduce the antimildew performance of bamboo treatment, and improve the control effect of Penicillium orange. In this study, the compounding process, bacteriostatic properties, and control efficacy of the citral treatment of bamboo were investigated, laying the foundations and providing a theoretical reference for the application of citral-based bamboo mold prevention.

## 2. Materials and Methods

### 2.1. Materials

Citral (97%), Tween 80, thymol (T1, ≥99%), cinnamaldehyde (98%), and all pure reagents were purchased from Sinopharm Chemical Reagent Co., Ltd (Shanghai, China) The bamboo test materials were procured from Moso bamboo. These materials were green and yellow strips processed into a longer bamboo piece and then cut into short bamboo pieces sized 50 mm × 20 mm × 5 mm (length × width × thickness). The bamboo pieces did not contain bamboo knots and their moisture content was approximately 10%. The bamboo strips were purchased from Xizhuyuan Bamboo Products Factory in Zhenghe County (Nanping, China) *Penicillium citrinum* (*PC*), *Trichoderma viride* (*TV*), *Aspergillus niger* (*AN*), and mixed mold (Hun, mixed in equal proportions of *PC*, *TV*, and *AN*) were the experimental strains used. 

### 2.2. Methods

#### 2.2.1. Preparation of Citral Compound

According to the experimental results of Zhang [16], the inhibition rates of 50 mg/mL of citral on *TV*, *AN*, and Hun exceeded 100%, but the bacteriostatic rate against *PC* was only 32.60%. However, when the citral concentration increased to 75 mg/mL, the inhibition rate of citral against *PC*, *TV*, *AN*, and Hun exceeded 100%. To this end, 75 mg/mL was used as the judgment concentration to verify the bacteriostatic properties of the citral compound. Citral was compounded with cinnamaldehyde (C_1_) or thymol (T_1_) in a 2:1 ratio, and the citral compound was prepared. The contents of each component are presented in Table 1. 

First, 5000 mg of citral was weighed in a beaker, and 2500 mg of cinnamaldehyde or thymol was added to the beaker, followed by the addition of Tween 80, which accounted for 2% of the total volume. Then, an appropriate amount of deionized water was added to the solution. At room temperature, the solution was stirred with a magnetic stirrer for 2 h. Finally, the solution was transferred into a volumetric flask, and deionized water was added to a fixed capacity of 100 mL, leading to the preparation of 75 mg/mL CC_1_ and CT_1_ citral compounds.

#### 2.2.2. Fourier-Transform Infrared Spectroscopy Analysis of Citral Compounds

Equal concentrations of citral (C), citral and cinnamaldehyde compound (CC_1_), and citral and thymol compound (CT_1_) were added dropwise to KBr flakes. Then, the molecular structure of the compound was characterized using the IR-Prestige-21 Fourier-transform infrared spectroscopy (FT-IR) instrument (Purchased from Shimadzu, Japan), and the molecular structure of the compound was characterized at a resolution of 4 cm^−1^ and a wavelength range of 500–4000 cm^−1^. Whether the molecular structure of the two substances in the compound changed was determined through infrared spectroscopy.

#### 2.2.3. Bacteriostatic Properties of Citral Compounds against Bamboo Mold

The bacteriostatic properties of 75 mg/mL citral compound on bamboo mold was explored using the Oxford cup method [37]. First, the sterilized Oxford cup (outer diameter: 8 mm; inner diameter: 6 mm) was placed on a medium plate (diameter: 90 mm) coated with 80 μL of bacterial solution. Then, 80 μL of different concentrations of the citral compound was injected into the cup. Second, the Petri dish was sealed with a sterile sealing membrane and placed at 4 °C for 2 h. Finally, the dish was placed in a constant temperature incubator at 28 °C with 85% ± 5% humidity. After 2 days, the diameter of the bacteriostatic ring was measured using the cross-cutting method, and the bacteriostatic rate was calculated using Formula (1). Each test was repeated 3 times, and the results were averaged. Tween 80 was considered as the control.
(1)$$\mathrm{Bacteriostatic}\;\mathrm{rate}=\frac{\begin{array}{c} \mathrm{Diameter}\;\mathrm{of}\;\mathrm{the}\;\mathrm{inhibition}\;\mathrm{zone}\;\mathrm{in}\;\mathrm{the}\;\mathrm{treatment}\;\mathrm{group}-\mathrm{diameter}\\ \mathrm{of}\;\mathrm{the}\;\mathrm{inhibition}\;\mathrm{zone}\;\mathrm{in}\;\mathrm{the}\;\mathrm{control}\;\mathrm{group} \end{array}}{\mathrm{Diameter}\;\mathrm{of}\;\mathrm{the}\;\mathrm{inhibition}\;\mathrm{zone}\;\mathrm{in}\;\mathrm{the}\;\mathrm{control}\;\mathrm{group}}\times 100\%$$

#### 2.2.4. Antimildew Treatment of Bamboo with Citral Compound

We found [17] that the control efficacy of the antimold treatment with citral reached 100% only when the citral concentration reached 200 mg/mL. Therefore, citral, citral cinnamaldehyde compound (CC_1_), and citraldehyde thymol compound (CT_1_) were prepared at a concentration of 200 mg/mL and placed in a sealed treatment tank. Then, the bamboo pieces were placed in the treatment tank for pressurized impregnation treatment. Bamboo pieces were subjected to pressure impregnation at 0.3 MPa after treatment for 90 min. Then, the treated bamboo pieces were taken out, the excess treatment agent on the surface was absorbed and weighed, and the drug load was calculated according to Formula (2).
(2)$$R=\frac{\left(m_{2}-m_{1}\right)\times \;C\;\times 10^{6}}{S}$$
where R is the drug load, g/m^2^; *m*_1_ is the mass of bamboo chips before treatment, g; *m*_2_ is the mass of bamboo chips after treatment, g; C is the concentration (mass fraction) of citral, citral cinnamaldehyde compound and citral thymol compound, %; *S* is the surface area of the bamboo sheet, mm^2^.

#### 2.2.5. Distribution of Citral Compound in Antimildew-Treated Bamboo

The compounded agent (CT_1_) of the citral and thymol compound was selected as the test material, and the distribution of this compound in the treated bamboo piece was qualitatively analyzed using the UV spectrophotometer (Purchased from Shimadzu, Japan). First, an equal mass of thymol and the compounding agent (Tween 80, citral, and thymol, and other qualities mixed) was mixed with absolute ethanol to prepare thymol and compound standard solutions, respectively. Second, the qualitative analysis of the treated bamboo pieces was performed in the direction of the length of the bamboo pieces at 0–1 mm (l_1_), 1–2 mm (l_2_), and 2–3 mm (l_3_), intercepted by a thin layer of bamboo (interception schematic diagram, Figure 1). The crushing equipment was used to prepare bamboo powder; then, 0.05 g of the powder was diluted with absolute ethanol to 20 mL. The solution was ultrasonicated for 15 min after centrifugation at 9000 r/min for 10 min. Then, 0.1 mL of the supernatant was mixed with absolute ethanol to fix the volume to 10 mL. The prepared thymol standard solution, the compound standard solution, and the antimildew-treated bamboo supernatants were placed in quartz cuvettes with a light path length of 1 cm and scanned using a UV-1800 ultraviolet spectrophotometer to detect their absorbances at a wavelength range of 200–400 nm. Absolute ethanol was used as a reference, and the scanning step size was 1 nm.

#### 2.2.6. Control Effect of the Citral Compound against Bamboo Mildew

With reference to the relevant provisions of the national standard “Test Method for the Prevention and Control of Wood Mold and Color-Changing Bacteria by Anti-mildew Agents” (GB/T 18261-2013) [38], antimildew experiments and tests were conducted to evaluate the control effectiveness of citral compounds against bamboo mildew. We observed and recorded the sizes of the area infected on the bamboo flakes in the incubator by *PC*, *TV*, *AN*, and Hun to determine the mold level of the bamboo flakes (Table 2). Then, the control effect on treated bamboo chips was calculated according to Formula (3).
(3)$$E=(1-\frac{D_{1}}{D_{0}})\times 100\%$$
where E is the effectiveness of prevention and control, %; *D*_1_ is the average infection value of bamboo tablets treated with the citral compound; *D*_0_ is the average infection value of bamboo tablets in the control group.

## 3. Results and Discussion

### 3.1. FT-IR Analysis of Citral and Its Compound Solutions

An appropriate amount of CC_1_, CT_1_, and citral solutions (control group) was used for FT-IR analysis, and the infrared spectra are presented in Figure 2.

As shown in Figure 2, 3445 cm^−1^ is the telescopic vibration of free-OH; 2920 cm^−1^ and 2860 cm^−1^ are the absorption peaks of saturation-CH_2_; 2722 cm^−1^ is the telescopic vibration of -CH in the aldehyde group; 1678 cm^−1^ is the absorption peak generated by C=O telescopic vibration; 1625 cm^−1^ is the absorption peak of C=C; 1580 cm^−1^ is the C=C in the benzene ring; 806 cm^−1^ and 690 cm^−1^ are the absorption peaks generated by extra-plane deformation vibration of C-H containing three C-H groups on the benzene ring; 748 cm^−1^ is the C-H extra-surface deformation vibration absorption peak in the monosubstituted benzene. Citral exhibited absorption peaks at 3445, 2920, 2860, 2722, 1678, and 1625 cm^−1^. This is mainly because the citral structure contains a saturated -CH2 group, one aldehyde group, and two C=C groups. This result is similar to that of previous studies [39,40]. Compared with citral, CC1 exhibited absorption peaks at 1580, 748, and 690 cm^−1^, mainly because cinnamaldehyde is a single-substituted derivative of benzene, and the expansion and contraction vibration of C=C and the off-plane deformation vibration of C-H occurred in the monosubstituted benzene. CC_1_ was enhanced at 1678 and 1625 cm^−1^, mainly because cinnamaldehyde also has an alkene aldehyde group [41]. However, the increased intensity of CC_1_ at 1678 was not twice as that of citral, mainly because a chemical reaction occurred after cinnamaldehyde and citral were mixed, which changed the structure of cinnamaldehyde and citral, thereby reducing the bacteriostatic properties of CC_1_.

CT_1_ was enhanced at 3445 cm^−1^ compared with citral, mainly due to the telescopic vibration absorption of -OH in thymol [42]. Moreover, 806 cm^−1^ and 690 cm^−1^ were the off-plane deformation vibration absorption peaks of C-H containing three C-H on the benzene ring; CT_1_ had absorption peaks in both places, mainly because thymol is a derivative of benzene with three substituents on the benzene ring. The absorption peak of CT_1_ at 1678 cm^−1^ was slightly reduced, mainly because the thymol structure does not contain C=O. Moreover, no absorption was observed, and a small part of thymol and citral possibly underwent a condensation reaction, consuming C=O in citral and leading to a decrease in the absorption peak of CT_1_. Therefore, a chemical reaction is believed to have occurred after the compounding of thymol and citral, changing the structure of some thymol and citral and thus altering the bacteriostatic properties.

In summary, compounding of citral and cinnamaldehyde and citral and thymol may have changed the structure of these compounds, which may have affected their bacteriostatic properties.

### 3.2. Bacteriostatic Properties of Citral Compounds

The inhibitory effects of CC_1_, CT_1_, and citral solutions (control group) on the common molds of bamboo, that is, *PC*, *TV*, *AN*, and Hun at a concentration of 75 mg/mL, are shown in Figure 3 and Figure 4, and Table 3.

As shown in Figure 3 and Figure 4, 75 mg/mL of CC_1_ generated relatively obvious bacteriostatic circles against *PC*, *TV*, *AN*, and Hun, with diameters of 14.13, 14.04, 13.91, and 13.29 mm, respectively; however, the diameters were 56%, 44%, 48%, and 58%, respectively, which are smaller than the diameters of the bacteriostatic circles with the same concentration of citral. Therefore, the bacteriostatic properties of the CC_1_ were weaker than those of citral. Figure 3 and Figure 4 also reveal that 75 mg/mL CT_1_ generated significantly larger bacteriostatic circles against the four types of molds than CC_1_. Compared with the same concentration of citral, CT_1_ generated larger bacteriostatic circles against *PC*, *AN*, and Hun, but the size of the bacteriostatic circle against *TV* was not much different between CT_1_ and citral. The diameters of the bacteriostatic circles generated with CT_1_ were 38.74, 30.51, 25.84, and 26.87 mm, which are 154%, 96%, 89%, and 117% larger than those generated by the citral treatment group, respectively.

Therefore, the bacteriostatic properties of CT_1_ against *TV*, *AN*, and Hun did not change significantly, but those against *PC* were significantly stronger than those of citral at the same concentration. Table 3 shows the inhibition rates of CC_1_ against *PC*, *TV*, *AN*, and Hun, which were 33.39%, 100.57%, 86.96%, and 91.50%, respectively.

The inhibition rates of CT_1_ against *PC*, *TV*, *AN*, and Hun were 269.30%, 335.86%, 247.31%, and 287.18%, respectively, which were 8.1, 3.3, 2.8, and 3.1 times those of CC_1_, respectively. The inhibition rates of CT_1_ against common molds in bamboo were considerably greater than those of CC_1_.

Qu [43] found that the antibacterial effect of cinnamon essential oil and mountain candy essential oil had a complex additive effect; the combination of cinnamon and mountain candy essential oil has a better antibacterial effect on BQM than the two used alone [44]. This shows that essential oils can play a synergistic effect after a certain volume ratio compounding and obtain a better antibacterial effect than a single essential oil. Wu [45] also concluded that blended plant essential oils exhibit additive effects.

Therefore, the selection of CT_1_ as an antimildew agent for bamboo mildew treatment can help achieve the goal of reducing the amount of citral while maintaining its strong bacteriostatic effect on bamboo molds.

### 3.3. Distribution of Citral Compound in Antimildew-Agent-Treated Bamboo

Figure 5 shows the absorption spectra of the standard solution (C) of citral standard solution (C), Tween 80 standard solution (T), citral and Tween 80 concentration mixed standard solution (CT), thymol standard solution (T_1_), and citral thymol compound standard solution in the wavelength range of 200–400 nm. The distribution of the standard solution of the citraldehyde thymol compound (CTT_1_) in bamboo is shown in Figure 6.

Figure 5 reveals that the thymol standard solution had two absorption peaks in the wavelength range of 200–400 nm, that is, at 223 nm and between 274 and 283 nm. Of them, the absorption peak between 274 and 283 nm was relatively flat. Compared with the thymol standard, the absorption wavelength at 223 nm changed and moved toward the longer wavelength, whereas the absorption peak at 274–283 nm remained unchanged. This may be because the absorption peak of the mixed solution such as citral and Tween 80 was at 233 nm, and when thymol with a weak polarity was added, it may have shifted and moved in the direction of short waves, resulting in the absorption peak of CT_1_ at 230 nm. Moreover, the absorption peak of the citral compound at 230 nm was considerably stronger than that of the thymol standard, and the absorption peak of the citral compound at 230 nm was speculated to have mainly been generated by the absorption of citral and Tween 80 in the compound. Therefore, the absorption strength of the solution between 274 and 283 nm can be used as the basis for judging the distribution of thymol in the antimildew-treated bamboo, and the absorption strength of the solution at 230 nm can be used as the basis for judging the distribution of citral in the bamboo treated with the CT_1_. Figure 6 shows that antimildew treatment with 200 mg/mL CT_1_ was not obvious in the length directions of 0–1 mm (l_1_), 1–2 mm (l_2_), and 2–3mm (l_3_) of the bamboo supernatant. The absorbance intensity at 230 nm gradually increased. Between 273 and 283 nm, the difference in the absorbance intensities of l_1_ and l_2_ was not obvious but was significantly lower than that of l_3_. This indicates that when the limonaldehyde thymol compound was dried on bamboo, it may have led to volatilization and loss of a part of the compound on the bamboo surface, thereby changing the distribution of CT_1_ on the surface. Thus, the limonaldehyde thymol compound showed a low distribution trend outside and vice versa inside in the treated bamboo, which is consistent with the conclusion drawn by Zhang [17].

### 3.4. Drug Loading of Citral Compound Agent in Antimildew-Treated Bamboo and Control Effect of Antimildew-Treated Bamboo

From Table 4, it can be seen that the drug load of citral-compound-treated-bamboo is slightly lower than that of citral-treated bamboo, which may be because the molecular weight of cinnamaldehyde [46] and thymol [47] is less than that of citral. In the process of pressurized impregnation of bamboo sheets, cinnamaldehyde or thymol with a small molecular weight in the compound is easier to immerse into the bamboo body, which occupies part of the bamboo void first, resulting in a corresponding decrease in the amount of citral immersion, so the drug load of the compound treatment bamboo is slightly smaller than that of the citral-treated bamboo. After 28 days of antimildew test, CC_1_ and CT_1_ antimildew treatment bamboo pieces have good control effect on common mold in bamboo. Among them, CT_1_ has a control effect of 100% on all molds; the control effect of CC_1_ against *PC*, *TV*, and Hun also reached 100%; however, the control effect of *AN* only reached 99.65%. Therefore, the antimildew treatment effect of CT_1_ was better than that of CC_1_. In addition, it can be seen from Table 4 that CT_1_ can achieve the same prevention and treatment efficacy compared with citral under a relatively small drug load. Therefore, the citral thymol compound CT_1_ is the best compound antifungal agent for bamboo mildew treatment.

### 3.5. Antimildew Properties of Citral Compounds

Figure 7 shows the antimildew treatment of bamboo with a concentration of 200mg/mL of citral thymol compound (CT_1_), and the antimildew effect of treated and untreated wood (control group) on the common *PC*, *TV*, *AN*, and Hun of bamboo on the 28th day.

As shown in Figure 7, the bamboo piece treated with CT_1_ did not show any growth of *PC*, *TV*, *AN*, or Hun on its surface on day 28. On the surface of the untreated bamboo, the four molds exhibited increased growth; *AN* and Hun almost covered the entire surface of the bamboo piece, causing severe damage to the bamboo piece. CT_1_ exhibited good antimildew properties, which were consistent with the antimildew effect of 200 mg/mL citral on day 28 in the experiment by Zhang [17]. Because 200 mg/mL CT_1_ comprises citral and thymol in a 2:1 ratio, the amount of citral used is reduced by approximately 67 mg/mL. This result is very close to the data of Ju [48] in their study of the synergistic bacteriostatic mechanism of citral against *AN*. Additionally, the antimildew performance of bamboo has not decreased.

Additionally, the antimildew performance of citral against bamboo is not reduced. Therefore, the CT_1_ preparation can prevent bamboo mildew. With CT_1_, the amount of citral required is reduced without any reduction in its antimildew performance against bamboo.

## 4. Conclusions

The antimildew treatment of bamboo with CT_1_ prepared by compounding citral and thymol at a ratio of 2:1 exhibited good antimildew properties. At the CT_1_ concentration of 200 mg/mL, the amount of citral required was reduced by approximately 67 mg/mL, without any reduction in its antimildew performance against bamboo. Compounding of citral with cinnamaldehyde or thymol may result in some chemical reactions, thereby changing the chemical structure of citral and affecting its bacteriostatic properties. The content of citral compound at a bamboo piece length of 0–3 mm increased with an increase in the depth of the material (the direction of the length of the bamboo sheets). Drying of the bamboo treated with the citral compound may have led to volatilization and loss of a part of the compound on the bamboo surface, thereby changing the distribution of CT_1_ on the surface. This showed that the distribution of CT_1_ was low distribution trend outside and vice versa inside in the treated bamboo. The experimental results provide a theoretical basis and technical reference for citral compound in bamboo mildew control.

## Figures and Tables

**Figure 1 polymers-14-04691-f001:**
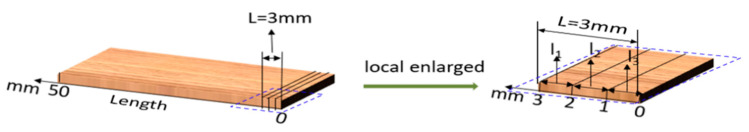
Screenshots of l_1_, l_2_, and l_3_.

**Figure 2 polymers-14-04691-f002:**
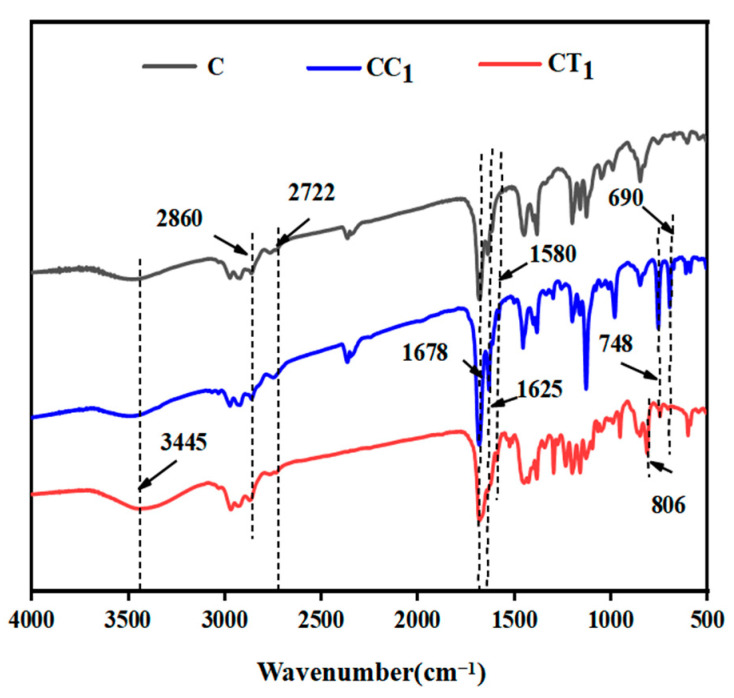
Infrared spectra of citral and citral compound solutions (C, citral solution; CC_1_, a compound formulated with citral and cinnamaldehyde; CT_1_, a compound formulated with citral and thymol).

**Figure 3 polymers-14-04691-f003:**
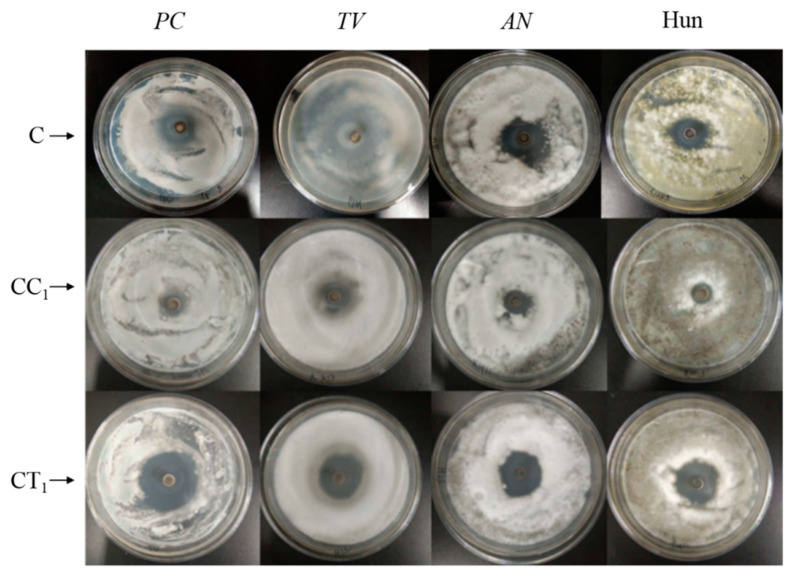
Inhibitory effect of citral and citral compound solutions against bamboo molds (C, citral solution; CC_1_, a compound formulated with citral and cinnamaldehyde; CT_1_, a compound formulated with citral and thymol).

**Figure 4 polymers-14-04691-f004:**
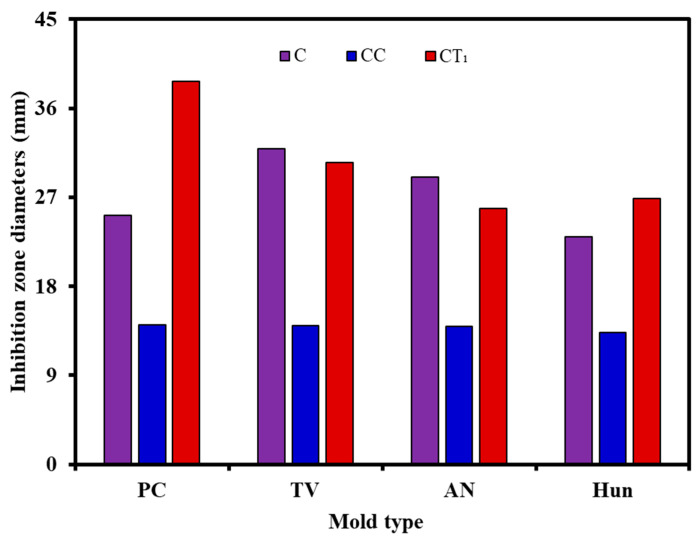
Effects of citral and citral compound solutions on the diameters of the inhibition zone of bamboo molds (C, citral solution; CC_1_, a compound formulated with citral and cinnamaldehyde; CT_1_, a compound formulated with citral and thymol).

**Figure 5 polymers-14-04691-f005:**
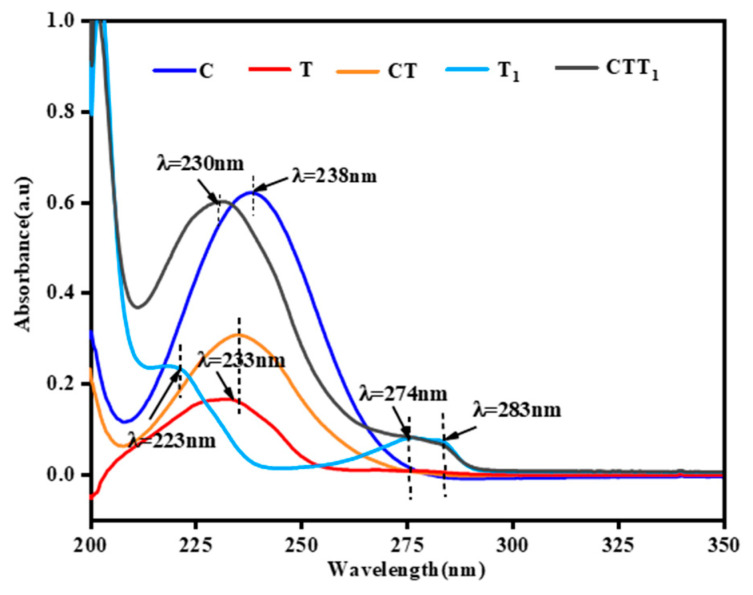
Ultraviolet absorption spectra of the standard solutions (C, citral standard solution; T: Tween 80 standard solution; CT, standard solution mixed of citral and Tween 80; T_1_, thymol standard solution; CTT_1_, standard solution mixed of citral and thymol).

**Figure 6 polymers-14-04691-f006:**
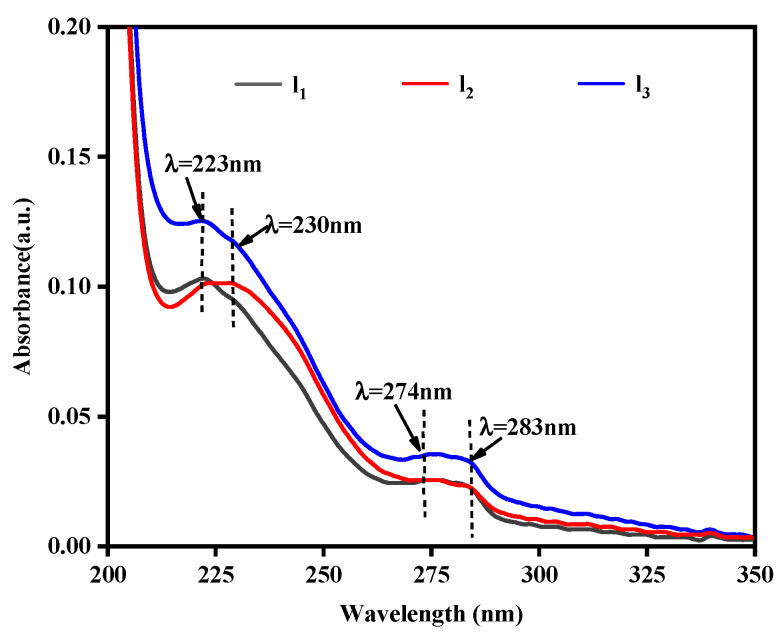
Ultraviolet absorption spectra of bamboo strips treated with citral and thymol compound (l_1_, supernatant of length 0–1 mm; l_2_, supernatant of length 1–2 mm; l_3_, supernatant of length 2–3 mm).

**Figure 7 polymers-14-04691-f007:**
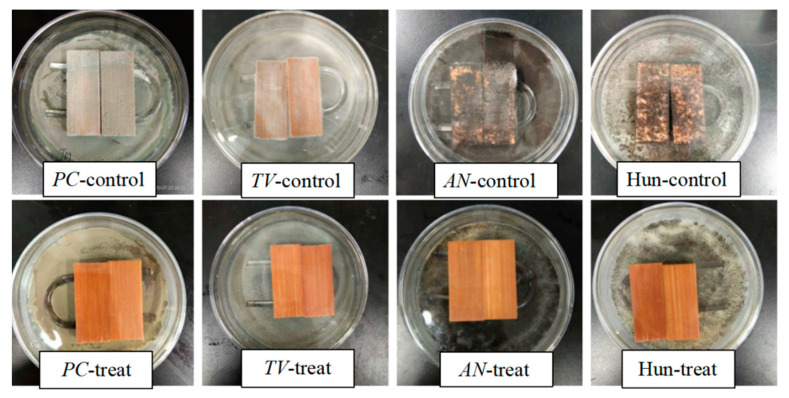
Antimildew photographs of untreated and treated bamboo strips on day 28.

**Table 1 polymers-14-04691-t001:** Amount of each main substance in the prepared citral compound.

Name of the Citral Compound	Quality of Citral/mg	Quality of Cinnamaldehyde/mg	Quality of Thymol/mg	Tween 80 Volume/mL
CC_1_ (citral + cinnamaldehyde)	5000	2500	–	2
CT_1_ (citral + thymol)	5000	–	2500	2

**Table 2 polymers-14-04691-t002:** Classification standard of surface infection levels of samples.

Infection Value	Specimen Infection Area
0	No hyphae or mildew on the sample surface
1	Infected area of sample < 1/4
2	Infected area of sample 1/4–1/2
3	Infected area of sample 1/2–3/4
4	Infected area of sample > 3/4

**Table 3 polymers-14-04691-t003:** Effects of citral and citral compound solutions on the inhibition rates of bamboo mold.

Name	The Bacteriostatic Rate of Bamboo Mold (%)			
	*PC*	*TV*	*AN*	Hun
C	140.06	356.28	290.12	231.12
CC_1_	33.39	100.57	86.96	91.50
CT_1_	269.30	335.86	247.31	287.18

Note: C, citral; CC_1_, citral cinnamaldehyde compound; CT_1_, citral thymol compound.

**Table 4 polymers-14-04691-t004:** The drug load of antimildew-treated bamboo with citral and citral compound and the control effect of antimildew treated bamboo on the 28th day (C, citral solution; CC_1_, a compound formulated with citral and cinnamaldehyde; CT_1_, a compound formulated with citral and thymol).

Name	Drug Loading (g/m^2^)	Antimildew Efficiency (%)			
		*PC*	*TV*	*AN*	Hun
C	119.250	100	100	100	100
CC_1_	108.219	100	100	99.65	100
CT_1_	116.270	100	100	100	100

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
