# Peer review of "Preparation of Citral Compound and Its Bamboo Antimildew Properties"

_polymers, 2022, doi:10.3390/polym14214691_

Round 1

Reviewer 1 Report

The authors presented anti-mildew effect of citral compound on bamboo. The work is acceptable for publication in Polymers. There are some language corrections for consideration e.g.:

"... pound showed a distribution trend of low outside and high inside in the treated bamboo" -> "... pound showed a low distribution trend outside and vise versa inside in the treated bamboo"

Please also check and polish the manuscript technically, and grammatically. 

Author Response

Response to comments of reviewers and editors

Reviewer1:

  1. The authors presented anti-mildew effect of citral compound on bamboo. The work is acceptable for publication in Polymers. There are some language corrections for consideration e.g.:"... pound showed a distribution trend of low outside and high inside in the treated bamboo" -> "... pound showed a low distribution trend outside and vise versa inside in the treated bamboo"Please also check and polish the manuscript technically, and grammatically.

Response: Thank you for your professional advice. We have revised the manuscript according to your valuable and helpful comment. At the same time, we checked the manuscript carefully and tried our best to improve the language.

Reviewer 2 Report

The paper has scientific merit but it requires an extensive modifications.

The introduction is not well-written, The authors should rewrite it by including more literature and better justification.

The methodology is OK. However, it seems that experimental results is lacking. How about the treatability of the bamboo? what is the weight percent gain after treatment? and how will that influence the anti-mildew performance? Do you perform leaching test? it is also important to test the efficacy of the treatment. 

Results and discussion - more comparison with previous studies are needed.

Conclusion - please rewrite the conclusion.

Author Response

  1. 1.The paper has scientific merit but it requires an extensive modifications.The introduction is not well-written, The authors should rewrite it by including more literature and better justification.

Response:Thank you for your professional advice. We have added more literature in the introduction to support my point. The revised introduction reads as follows:

Bamboo, a natural material, has the tensile strength comparable to that of low-carbon steel and good toughness and degradability, and it is highly advantageous in terms of structure and cost[1]. Therefore, bamboo and its products have been widely used for activities such as decoration, architecture, and gardening[2-5] and exhibit a rapid development trend. However, bamboo and its products are very susceptible to mildew and severe surface pollution, and thus, they lose their use value[6,7]and are discarded, which results in considerable wastage of resources and economic losses. Therefore, vigorous bamboo mold prevention research is of great significance.

An anti-mildew agent for bamboo has recently become a hot research topic, and the research and application of natural antibacterial agents for bamboo mold prevention have been increasing. Citral is one such natural antibacterial agent and is mainly derived from the essential oil of the Chinese medicine Shancangzi[8]. It is a fat-soluble natural mixture with rich fragrance[8] and has received more attention because of its broad-spectrum and strong bacteriostatic properties[9,10].Studies have shown that citral has good antibacterial properties against Staphylococcus aureus, Escherichia coli, Penicillium and Mucormyces[11]; Citral and its derivatives have a certain inhibitory effect on Camellia oleifera anthracnose, and the inhibitory effect increases with the increase of concentration[12].Xia[13]studied the inhibitory effect of citral on Aspergillus flavus, the source of Chinese herbal medicines, and the results showed that citral had a good control effect on mildew in Chinese herbal medicines, and the concentrations of Ze diarrhea and citrus aurantium mildew were 4μL/g and 8μL/g, respectively.Tao[14,15]studied the antifungal activity and antifungal mechanism of citrus essential oil and citral against Penicillium italicans and Penicillium dactylidae, and found that citral can significantly change the morphology of Penicillium italicans mycelium by causing cytoplasmic loss and deformation of hyphae.However, no one has studied whether citral has an inhibitory effect on bamboo mold, so the author's team [16,17] conducted a study on the bacteriostatic performance of citral on common molds in bamboo and its control effect on bamboo treatment.The results showed that when the citral concentration was 75 mg/mL, the bacteriostatic rate of citral on bamboo molds exceeded 100%. However, only when the citral concentration reached 200 mg/mL, the control effectiveness of citral against the bamboo mildews reached 100%. The good anti-mildew performance of citral can be achieved only by increasing the citral concentration.However, when the amount of citral is high, the lemon flavor is strong, which has a negative impact on the sensory characteristics of the product itself and its odor may cause discomfort to consumers. Thus, reducing the amount of citral in bamboo mold prevention is essential. However, this would reduce the antimildew performance of citral, which necessitates studies on the process and method of reducing the amount of citral without reducing its anti-mildew performance.

We found[16] that the citral treatment of bamboo had a poor control effect on Penicillium citrinum but a better control effect on Aspergillus niger, Trichoderma viride molds, and mixed molds. Studies have shown that the natural antibacterial agent cinnamaldehyde has a cinnamon smell and sweetness, is the main ingredient in the traditional Chinese medicine cinnamon branch and cinnamon[18] , and has a broad spectrum of antifungal effects [19-23]. Thymol has the spicy and herbaceous aromas of thyme, is the main component of plant volatile oils (about 50%) [24-27], and has a broad-spectrum antifungal effect [28-30]. Both thymol and cinnamaldehyde are listed as "generally recognized as safe" substances (GRAS) by the US Food and Drug Administration[31-33] , which can be used as green and safe natural antibacterial agents in food preservation, pharmaceutical preparations and feed preservatives, in addition, studies have shown that cinnamaldehyde and thymol[34-36]have a good antibacterial effect on Penicillium orange. Therefore, in view of the characteristics of cinnamaldehyde and thymol having the characteristics of green, safe and good antibacterial effect on Penicillium orange, the author selected cinnamaldehyde and thymol as the compound agent of citral for compounding, and through the synergistic effect between them, it can reduce the amount of citral use, do not reduce the anti-mildew performance of bamboo treatment, and improve the control effect of Penicillium orange.In this study, the compounding process, bacteriostatic properties, and control efficacy of the citral treatment of bamboo were investigated, laying the foundation and providing a theoretical reference for the application of citral to bamboo mold prevention.

2.The methodology is OK. However, it seems that experimental results is lacking. How about the treatability of the bamboo?what is the weight percent gain after treatment? and how will that influence the anti-mildew performance? Do you perform leaching test? it is also important to test the efficacy of the treatment. 

Response:Thank you very much for your careful review.

 In the early stage, according to the results of the study that the anti-mildew load of bamboo could not be achieved after treating bamboo chips with a minimum bacteriostatic concentration of 3 times, and the antibacterial rate of four kinds of bamboo mold was more than 100% with a citral concentration of 75mg/mL, 100mg/mL, 125mg/mL, 150mg/mL, 175mg/mL, and 200mg/mL. The effect of citral concentration on the wet drug load of anti-mildew treated bamboo tablets was discussed. The experiments showed that with the increase of citral concentration, the drug load of citral impregnated bamboo tablets with different concentrations gradually increased, reaching 51.620g/m2, 69.204g/m2, 71.566g/m2, 71.574g/m2, 84.457g/m2 and 119.250g/m2, respectively. Increasing the citral concentration of anti-mildew treated bamboo chips is still the most effective way to increase the drug load of anti-mildew bamboo tablets, thereby improving the control effectiveness of anti-mildew bamboo.

This article is based on the previous research, through the compound of citral with thymol and cinnamaldehyde to achieve the purpose of reducing the amount of citral without reducing the mildew resistance of bamboo treatment. Therefore, preliminary experiments on drug loading rates have been completed and will not be discussed in this article; The results of the anti-attrition experiment will be reported by the authors in a separate article.

 3.Results and discussion - more comparison with previous studies are needed.

Response: Thank you for your professional advice, we have compared with previous studies in the results and discussion section, and some literature is added here to support it.

Mainly because the citral structure contains saturated -CH2 group, one aldehyde group, and two C=C groups.This result is similar to previous studies[39-40].

Compared with citral, CC1 exhibited absorption peaks at 1580, 748, and 690 cm−1, mainly because cinnamaldehyde is a single-substituted derivative of benzene, and the expansion and contraction vibration of C=C and the off-plane deformation vibration of C-H occurred in the monosubstituted benzene. CC1 was enhanced at 1678 and 1625 cm−1, mainly because cinnamaldehyde also has alkenealdehyde group[41].

CT1 was enhanced at 3445 cm−1 compared with citral, mainly due to the telescopic vibration absorption of -OH in thymol[42].

Qu[43]found that the antibacterial effect of cinnamon essential oil and mountain candy essential oil had a complex additive effect; The combination of cinnamon and mountain candy essential oil has a better antibacterial effect on BQM than the two used alone[44]; This shows that essential oils can play a synergistic effect after a certain volume ratio compounding, and obtain a better antibacterial effect than a single essential oil. Wu[45] also concluded that blended plant essential oils exhibit additive effects.

This result is very close to the data of Ju[46] in the study of the synergistic bacteriostatic mechanism of citral against Aspergillus niger. And the anti-mildew performance of bamboo has not decreased.

4.Conclusion - please rewrite the conclusion.

Response:Thank you for your professional advice. We have rewritten the conclusion as follows:

The anti-mildew treatment of bamboo with CT1 prepared by compounding citral and thymol at a ratio of 2:1 exhibited good anti-mildew properties. At the CT1 concentration of 200 mg/mL, the amount of citral required was reduced by approximately 67 mg/mL, without any reduction in its anti-mildew performance against bamboo. Compounding of citral with cinnamaldehyde or thymol may result in some chemical reactions, thereby changing the chemical structure of citral and affecting its bacteriostatic properties.The content of citral compound at the length of bamboo piece 0–3 mm increased with an increase in the depth of the material (the direction of the length of the bamboo sheets). Drying of the bamboo treated with the citral compound may have led to volatilization and loss of a part of the compound on the bamboo surface, thereby changing the distribution of CT1 on the surface. This showed that the distribution of CT1 was low distribution trend outside and vise versa inside in the treated bamboo.The experimental results provide a theoretical basis and technical reference for citral compound in bamboo mildew control.

Round 2

Reviewer 2 Report

The Introduction and Results and Discussion has been greatly improved. However, I am still sticking with my previous comments that the experiments are lacking to warrant publication.

Author Response

A:We have made changes to the article as requested by you.

2.2.4. Anti-mildew treatment of bamboo with citral compound

We found[17] that only when the citral concentration reached 200 mg/mL, the control

efficacy of the antimold treatment with citral reached 100%. Therefore, citral, citral

cinnamaldehyde compound (CC1), and citraldehyde thymol compound (CT1) were prepared

at a concentration of 200 mg/mL and placed in a sealed treatment tank. Then, the bamboo

pieces were placed in the treatment tank for pressurized impregnation treatment. Bamboo

pieces were subjected to pressure impregnation at 0.3 MPa after treatment for 90 min. Then,

the treated bamboo pieces were taken out, the excess treatment agent on the surface was

absorbed, weighed, and the drug load was calculated according to formula 2.

In the formula: R is the drug load, g/m2; m1 is the mass of bamboo chips before treatment, g;

m2 is the mass of bamboo chips after treatment, g; C is the concentration (mass fraction) of

citral, citral cinnamaldehyde compound and citral thymol compound, %; S is the surface

area of the bamboo sheet, mm2 .

3.4. Drug loading of citral compound agent in anti-mildew treated bamboo and control effect of

anti-mildew treated bamboo

Table 4. The drug load of anti-mildew treated bamboo with citral and citral compound and the control

effect of anti-mildew treated bamboo on the 28th day (C: citral solution; CC1: a compound formulated with

citral and cinnamaldehyde; CT1: a compound formulated with citral and thymol).

From Table 4, it can be seen that the drug load of citral compound treatment bamboo is

slightly lower than that of citral-treated bamboo, which may be because the molecular

weight of cinnamaldehyde [46] and thymol [47] is less than that of citralde. In the process of

pressurized impregnation of bamboo sheets, cinnamaldehyde or thymol with a small

Name

Drug loading(

g/m2)

Antimildew efficiency (%)

PC

TV

AN

Hun

C

119.250

100

100

100

100

CC1

108.219

100

100

99.65

100

CT1

116.270

100

100

100

100molecular weight in the compound is easier to immerse into the bamboo body, which

occupies part of the bamboo void first, resulting in a corresponding decrease in the amount

of citral immersion, so the drug load of the compound treatment bamboo is slightly smaller

than that of the citral-treated bamboo. After 28 days of anti-mildew test, CC1 and CT1

anti-mildew treatment bamboo pieces have good control effect on common mold in bamboo.

Among them, CT1 has a control effect of 100% on all molds, the control effect of CC1 against

PC, TV and Hun also reached 100%, but the control effect of AN only reached 99.65%, so

the anti-mildew treatment effect of CT1 was better than that of CC1. In addition, it is also

known from Table 4 that CT1 can achieve the same prevention and treatment efficacy

compared with citral under a relatively small drug load. Therefore, citral thymol compound

CT1 is the best compound antifungal agent for bamboo mildew treatment.
